# Risk of major autoimmune diseases in female breast cancer patients: A nationwide, population-based cohort study

Hsin-Hua Chen[1,2,3,4,5,6], Ching-Heng Lin[2], Der-Yuan Chen[7,8,9], Wen-Cheng Chao[2], Yi-Hsing Chen[3,10], Wei-Ting Hung[10], Yin-Yi Chou[10], Yi-Da Wu[10], Chien-Chih Chen[1,11]*

1 Ph.D. Program in Translational Medicine, National Chung-Hsing University, Taichung, Taiwan, Republic of China, 2 Department of Medical Research, Taichung Veterans General Hospital, Taichung, Taiwan, Republic of China, 3 School of Medicine, National Yang-Ming University, Taipei, Taiwan, Republic of China, 4 Institute of Biomedical Science and Rong Hsing Research Center for Translational Medicine, Chung-Hsing University, Taichung, Taiwan, Republic of China, 5 Department of Industrial Engineering and Enterprise Information, Tunghai University, Taichung, Taiwan, Republic of China, 6 Institute of Public Health and Community Medicine Research Center, National Yang-Ming University, Taipei, Taiwan, Republic of China, 7 Rheumatology and Immunology Center, China Medical University Hospital, Taichung, Taiwan, Republic of China, 8 Translational Medicine Laboratory, Rheumatology and Immunology Center, China Medical University Hospital, Taichung, Taiwan, Republic of China, 9 School of Medicine, China Medical University, Taichung, Taiwan, Republic of China, 10 Division of Allergy, Immunology and Rheumatology, Department of Internal Medicine, Taichung Veterans General Hospital, Taichung, Taiwan, Republic of China, 11 Department of Radiation Oncology, Taichung Veterans General Hospital, Taichung, Taiwan, Republic of China

* chiencheh@gmail.com

**Data Availability Statement:** All relevant data are within the paper.

## Abstract

### Background

Breast cancer is one of the most common malignancies among women. However, there remains no consensus in current literature on the incidence of autoimmune diseases among breast cancer patients. The purpose of this study was to evaluate the risks of major autoimmune diseases (MAD) including systemic lupus erythematosus (SLE), rheumatoid arthritis (RA), Sjögren's syndrome (SS) and dermatomyositis (DMtis)/polymyositis (PM) in female breast cancer patients.

### Methods

Using the Taiwanese National Health Insurance Research Database (NHIRD) records from 2003 to 2013, we identified newly-diagnosed female breast cancer patients and randomly selected females without breast cancer in the period 2007 to 2013 into a control group. We matched the two cohorts using a 1:4 ratio based on age, and the year of index date for comparison of the risk of major autoimmune diseases. We estimated and compared the relative risks of autoimmune diseases in female breast cancer patients and females without breast cancer.

### Results

A total of 54,311 females with breast cancer and 217,244 matched females without breast cancer were included in this study. For SLE, the incidence rates were 2.3 (breast cancer

**Funding:** This study was supported by Taichung Veterans General Hospital grant CE16241B to H-HC.

**Competing interests:** The authors have declared that no competing interests exist.

**Abbreviations:** SLE, systemic lupus erythematosus; RA, rheumatoid arthritis; SS, Sjögren's syndrome; DMtis, dermatomyositis; PM, polymyositis; MADs, major autoimmune diseases; CTLA-4, cytotoxic T-lymphocyte–associated antigen 4; NHIRD, National Health Insurance Research Database; BCFs, breast cancer females; NBCFs, non-breast cancer females; NHI, National Health Insurance; LHID, Longitudinal Health Insurance Database; CIC, catastrophic illness certificate; ICD-9-CM, International Classification of Diseases, Ninth Revision, Clinical Modification; IRRs, incidence rate ratios..

group) vs. 10.0 (control group) per 100,000 women years; for RA rates were 19.3 (breast cancer group) vs. 42.7 (control group) per 100,000 women years; and for SS rates were 20.5 (breast cancer group) vs. 38.2 (control group) per 100,000 women years. After adjusting for potential confounders, the hazard ratios (95% confidence intervals) for female breast cancer patients vs. control group were 0.04 (0.01–0.24) for SLE; 0.03 (0.02–0.04) for RA; and 0.21 (0.09–0.48) for SS.

## Conclusion

Female breast cancer patients had lower risks of SLE, RA and SS when compared to female individuals without breast cancer. However, there was no significant difference in the risk of developing DMtis/PM between both groups.

## Introduction

Breast cancer is one of the most common malignancies among women. In Taiwan, autoimmune diseases were three to four times more common in women than in men [1]. However, there remains no consensus on the incidence rates of autoimmune diseases among breast cancer patients.

Previous studies [2–4] have demonstrated that breast cancer patients showed an increased spontaneous expression of cytotoxic T-lymphocyte–associated antigen 4 (CTLA-4) on T lymphocytes in breast tissue and peripheral blood mononuclear cells, leading to impaired T cell activation. T cell activation is involved in the development of autoimmune diseases such as rheumatoid arthritis [5], systemic lupus erythematosus [6–7], Sjögren's syndrome [8–9], dermatomyositis (DMtis) [10–11] and polymyositis (PM) [10–11]. Abatacept is a fusion protein, which links the extracellular domain of CTLA-4 to the Fc region of immunoglobulin IgG1. Abatacept, which works by interfering with the immune activity of T cells, has been used in the treatment of various autoimmune diseases [12–15], and is approved for RA therapy. DMtis/PM has been found to be associated with various cancers [16–19], including breast cancer [20]. Therefore, we hypothesized that breast cancer patients may have a lower risk of major autoimmune diseases, except DMtis and PM.

The Taiwanese National Health Insurance Research Database (NHIRD) is available for research purposes and has facilitated nationwide, population-based longitudinal studies. The aim of this study was to evaluate the risk of major autoimmune diseases in female breast cancer patients using the NHIRD.

## Methods

### Ethics statement

The study was permitted by the Institutional Review Board (IRB) of the Taichung Veterans General Hospital (IRB number: CE17100B). The requirement for informed consent was waived, since personal information was anonymized.

### Study design

This study was a retrospective cohort study.

## Data source

The data set were extracted from the NHIRD database using data from the period 2003 to 2013. In 1995, Taiwan initiated a compulsory National Health Insurance (NHI) program. The NHI program currently covers over 99% of the Taiwanese population. The NHIRD includes comprehensive data regarding medication prescription, ambulatory care services, admission services and traditional medical services. Some personal data and history such as body weight, body length, alcohol use, and smoking, are not available in the NHIRD. By updating original medical records regularly, the Bureau of NHI has improved the accuracy of data in the NHIRD [21]. The NHIRD is managed by the National Health Research Institute (NHRI). The NHRI provides research data from the database to researchers after anonymizing all personal information.

We utilized multiple NHIRD datasets in this study, including data on both inpatients and outpatients, as well as enrollment data from 2003 to 2013. In 2000, the NHRI randomly selected and enrolled one million representative individuals from the NHIRD, establishing the Longitudinal Health Insurance Database (LHID2000). Since we were unable to get all the data in NHIRD, we selected the non-breast cancer control group from the LHID cohort. We used LHID2000 claims data from 2003 to 2013 for analysis of the comparison cohort.

The Bureau of NHI also established a registry of patients with possible fatal illnesses, such as malignancies, SLE, RA, SS and DMtis/PM. Patients with a possible fatal illness were issued a catastrophic illness certificate and were exempt from copayment for all medications prescribed for a corresponding possible fatal illness diagnosis. However, a catastrophic illness certificate was issued only after a thorough review and validation of original medical records by two or more qualified specialists.

## Definition of female breast cancer patients

Female breast cancer patients were defined as having a catastrophic illness certificate for breast cancer [International Classification of Diseases, Ninth Revision, Clinical Modification (ICD-9-CM) code 174].

## Definition of major autoimmune diseases

Major autoimmune diseases in this study included SLE, RA, SS and DMtis/PM. Patients with major autoimmune diseases were defined as having a catastrophic illness certificate for SLE (ICD-9-CM code 710.0), RA (ICD-9-CM code 714.x), SS (ICD-9-CM code 710.2) or DMtis (ICD-9-CM code 710.3)/PM (ICD-9-CM code 710.4).

## Study outcome

The study outcome was the time from the index date to the time of the first diagnosis for major autoimmune diseases. We defined the censor date as December 31, 2013 (the last date of the database used) or the date of withdrawal from the NHI for emigration or death.

## Study subjects

**Newly-diagnosed female breast cancer patients from entire Taiwanese population.**
Using NHIRD information from 2003 to 2013, we identified 67,552 newly-diagnosed female breast cancer patients between 2007 and 2013 as the female breast cancer cohort. The index date for female breast cancer patients was defined as the first date of application for a female breast cancer catastrophic illness certificate. After excluding patients with a history of an ambulatory or inpatient visit resulting in diagnosis of any autoimmune disease including SLE

(ICD-9-CM 710.1), RA (ICD-9-CM 714.0), SS (ICD-9-CM 710.2), DMtis (ICD-9-CM 710.3) and PM (ICD-9-CM 710.4), we included 62,570 newly-diagnosed female breast cancer patients as our female breast cancer cohort.

**Matched non-breast cancer female individuals selected from a one million representative population.** Using LHID2000 information from 2003 to 2013, we identified 430,530 female individuals without breast cancer as the control group. After excluding patients with a history of an ambulatory or inpatient visit resulting in diagnosis of any autoimmune disease including SLE (ICD-9-CM 710.1), RA (ICD-9-CM 714.0), SS (ICD-9-CM 710.2), DMtis (ICD-9-CM 710.3) and PM (ICD-9-CM 710.4), 409,340 female non-breast cancer individuals were included before matching.

**Matching female breast cancer patients and female non-breast cancer individuals with a 1:4 ratio.** We recorded baseline differences between the female breast cancer cohort and the female non-breast cancer individuals to conduct propensity score matching (1:4). We used a multivariable logistic regression model to calculate the propensity score. The baseline variables for calculating the propensity score included age and year of index date (index year). We used the date of the first ambulatory or inpatient visit for any reason in the index year as the index date for female non-breast cancer individuals. Those who had a major autoimmune disease diagnosis before the index date were excluded. Finally, we included 54,311 female breast cancer patients and 217,244 matched female non-breast cancer individuals as the study subjects.

## Potential confounders

Potential confounders included age, Charlson comorbidity index (CCI) and concomitant medications including hormone therapy (i.e., Tamoxifen, anastrozole and letrozole) and chemotherapy (i.e., cyclophosphamide, doxorubicin, docetaxel, paclitaxel, methotrexate, fluorouracil, cisplatin, carboplatin). The CCI, as adapted by Deyo et al. [22], was used to represent the level of general comorbid medical conditions. For female breast cancer patients, breast cancer was not included for calculating CCI. The presence of a comorbidity was defined as having at least three ambulatory visits or one inpatient visit with a corresponding ICD-9-CM code within 1 year before the index date. Use of concomitant medication was defined as prescription of corresponding medications during the follow-up period.

## Statistical analysis

Continuous variables are shown as a mean ± standard deviation and categorical variables as a percentage of subjects. We tested the differences between continuous variables by using Student's $t$-test and between categorical variables by using Pearson's $\chi^2$ test. We estimated incidence rate ratios (IRRs) with 95% confidence intervals (CIs) to compare the incidences of major autoimmune diseases between groups. We quantified the associations between breast cancer and the development of SLE, RA, SS, DMtis /PM by estimating hazard ratios (HRs) with 95% CIs using Cox regression analysis adjusting for age, CCI, hormone therapy and chemotherapy. Statistical analyses were performed using SAS statistical software, version 9.3 (SAS Institute, Inc., Cary, NC, USA). A p-value less than 0.05 was considered statistically significant.

## Results

A total of 54,311 female breast cancer patients and 217,244 matched female non-breast cancer individuals were included. The mean ± SD age was 53.6 ± 12.7 years in both groups. Patients' characteristics are shown in Table 1.

The incidence rates of autoimmune diseases in both the female breast cancer cohort and female non-breast cancer cohort are shown in Table 2. Among female breast cancer patients,

**Table 1. Characteristics of female breast cancer patients and matched female non-breast cancer individuals.**

| | Female Non-breast cancer individuals | Female breast cancer patients | |
| --- | --- | --- | --- |
| | (n = 217,244) (%) | (n = 54,311) (%) | P-value |
| **Age**, years (mean ± SD) | 53.6 ± 12.7 | 53.6 ±1 2.7 | 1.000 |
| **CCI** (mean ± SD) | 0.3 ± 0.9 | 1.1 ± 2.1 | <0.001 |
| **CCI group** | | | |
| 0 | 176,864 (81.4) | 37,840 (69.7) | <0.001 |
| ≥1 | 40,380 (18.6) | 16,471 (30.3) | |
| **Concomitant medication** | | | |
| In the period from the index date to first SLE diagnosis date or censor date | | | |
| Hormone therapy# | 195 (0.1) | 36,789 (67.7) | <0.001 |
| Chemotherapy* | 2,898 (1.3) | 35,804 (65.9) | <0.001 |
| In the period form the index date to first RA diagnosis date or censor date | | | |
| Hormone therapy# | 195 (0.1) | 36,787 (67.7) | <0.001 |
| Chemotherapy* | 2,837 (1.3) | 35,802 (65.9) | <0.001 |
| In the period from the index date to first SS diagnosis or censor date | | | |
| Hormone therapy# | 195 (0.1) | 36,788 (67.7) | <0.001 |
| Chemotherapy* | 2,897 (1.3) | 35,803 (65.9) | <0.001 |
| In the period from the index date to first PM/DMtis diagnosis date or censor date | | | |
| Hormone therapy# | 195 (0.1) | 36,789 (67.7) | <0.001 |
| Chemotherapy* | 2,912 (1.3) | 35,804 (65.9) | <0.001 |

Data are shown as number (%) unless specified otherwise.

CCI, Charlson comorbidity index

#Hormone therapy included Tamoxifen, Anastrozole, and Letrozole

*Chemotherapy included Cyclophosphamide, Doxorubicin, Docetaxel, Paclitaxel, Methotrexate, Fluorouracil, Cisplatin, and Carboplatin

**Table 2. Incidence rates and risks of SLE, RA, SS and PM/DM in breast cancer females compared with matched female non-breast cancer individuals.**

| | SLE | RA | SS | DMtis/PM |
| --- | --- | --- | --- | --- |
| **Female non-breast cancer individuals (n = 217,244)** | | | | |
| Event (%) | 82 (0.04) | 348 (0.16) | 312 (0.14) | 14 (0.006) |
| Person-years | 816,360 | 815,557 | 815,713 | 816,581 |
| IR, /$10^5$ years | 10.0 | 42.7 | 38.2 | 1.7 |
| **Breast cancer females (n = 54,311)** | | | | |
| Event (%) | 4 (0.01) | 33 (0.06) | 35 (0.06) | 4 (0.01) |
| Person-years | 170,627 | 170,560 | 170,561 | 170,627 |
| IR, /$10^5$ years | 2.3 | 19.3 | 20.5 | 2.3 |
| **Breast cancer females compared with female non-breast cancer individuals** | | | | |
| IRR (95% CI) | 0.23 (0.09–0.64) | 0.45 (0.32–0.65) | 0.54 (0.38–0.76) | 1.37 (0.45–4.15) |
| HR# (95% CI) | | | | |
| Crude | 0.23 (0.08–0.62) | 0.45 (0.32–0.64) | 0.55 (0.39–0.78) | 1.35 (0.44–4.10) |
| Adjusted* | 0.04 (0.01–0.24) | 0.03 (0.02–0.04) | 0.21 (0.09–0.48) | 0.37 (0.08–1.80) |

Abbreviations: SLE, systemic lupus erythematosus; RA, rheumatoid arthritis; SS, Sjögren's syndrome; DMtis, dermatomyositis; PM, polymyositis; IRR, incidence rate ratio; CI, confidence interval; HR, hazard ratio

#Using Cox proportional hazard regression model

*Adjusted variables included age, Charlson comorbidity index, hormone therapy and chemotherapy

the incidence rate of SLE was 2.3 per 100,000 women years, the incidence rate of RA was 19.3 per 100,000 women years, the incidence rate of SS was 20.5 per 100,000 women years and the incidence rate of DMtis/PM was 2.3 per 100,000 women years. In the female non-breast cancer cohort, the incidence rate of SLE was 10.0 per 100,000 women years, the incidence rate of RA was 42.7 per 100,000 women years, the incidence rate of SS was 38.2 per 100,000 women years and the incidence rate of DMtis/PM was 1.7 per 100,000 women years. After adjusting for age, CCI, hormone therapy and chemotherapy, the HZs with corresponding 95% CIs for females breast cancer patients vs. control group were 0.04 (0.01–0.24) for SLE; 0.03 (0.02–0.04) for RA; and 0.21 (0.09–0.48) for SS. However, the risk of DMtis/PM was not significantly different between the female breast cancer group and the control group (HR, 0.37; 95% CI, 0.08–1.80).

## Discussion

This nationwide, population-based, matched cohort study was the first to investigate the risks of developing major autoimmune diseases including SLE, RA, SS, DMtis /PM, among female breast cancer patients. After adjusting for age, CCI, hormone therapy and chemotherapy, we found that patients with female breast cancer had a significant lower risk of developing SLE, RA, and SS, but not DMtis/PM compared to the control group.

The mechanism of lower autoimmune disease incidence rates in female breast cancer patients remains unknown. Jaberipour et al. [2] showed higher CTLA-4 expression and higher activity of regulatory T (Treg) cells in the peripheral blood mononuclear cells of female breast cancer patients. Mao et al. [3] demonstrated abnormal expression and dysregulation of CTLA-4 in female breast cancer patients. Khalife et al. [4] found overexpression of Treg-related markers and higher activity of Treg cells in the peripheral blood mononuclear cells of female breast cancer patients. Treg cells play an important role in the regulation of immune activities and maintain tolerance to self-antigens by suppressing antigen-presenting cells via CTLA-4. Overexpression of Treg-related markers and higher activity of Treg cells affect the immune system in female breast cancer patients. Therefore, we hypothesized that female breast cancer patients may have a decreased risk of autoimmune diseases and our results showed female breast cancer patients had a significant lower risk of developing SLE, RA and SS. On the other hand, several clinical immunotherapy studies [23–24] used Ipilimumab, a monoclonal antibody which targeted to CTLA-4 and activated the immune system, and showed rheumatic and musculoskeletal adverse effects in their trials.

DMtis and PM are chronic inflammatory diseases, and DMtis/PM patients have skeletal, muscle and skin symptoms. After reviewing DMtis/PM patients, Fang et al. [20] demonstrated a high frequency of malignancies (17.2%), and found the most common malignancies were nasopharyngeal cancer and female breast cancer. The cause of DMtis/PM is not clear. However, pathology studies have shown autoantibody in muscle biopsies, as discussed in previous studies [25–26]. According to Engel et al. [27], muscle biopsies showed CD4+ and CD8+ T cells and B cells in DMtis/PM patients. Of the other autoantibodies discussed in recent studies [28–29], the most common in DMtis and PM patients were anti-SSA, anti-Ro-52 and anti-PMScl [30–31]. Shah et al. [32] demonstrated that DMtis-associated autoantibodies were detectable in female breast cancer patients who had clinically specific rheumatic diseases. These cancer-associated autoantibodies may explain why the incidence rates of DMtis/PM were not significantly different between the female breast cancer cohort and the female non-breast cancer cohort.

This study has some limitations that should be addressed. Firstly, the accuracy of diagnoses based on data from NHIRD was of concern. However, the use of the catastrophic illness certificate to confirm the diagnosis of breast cancer and MADs may have increased the accuracy of

diagnoses. Secondly, some potential confounders including smoking, drinking and over-the-counter medications were not included in the claims data. Thirdly, treatment options varied depending on the particular disease, patients' condition, physicians' discretion and period of diagnosis. All of these could have affected individual immune responses, the incidence rates and risks of developing autoimmune diseases. Finally, these results may not be generalized to non-Taiwanese populations. In order to eliminate confounding factors, we adjusted for variables, including age, CCI, hormone therapy and chemotherapy. Further studies are still required to clarify the mechanisms of different autoimmune diseases in breast cancer patients.

## Conclusion

This nationwide, population-based cohort study showed that female breast cancer patients had lower risks of SLE, RA and SS when compared to female individuals without breast cancer. However, there was no significant difference in the risk of developing DMtis/PM between both groups. Further studies are warranted to clarify the mechanisms of these diseases in female breast cancer patients.

## Acknowledgments

This study was supported by Taichung Veterans General Hospital.

## Author Contributions

**Data curation:** Hsin-Hua Chen, Ching-Heng Lin, Wen-Cheng Chao, Yi-Hsing Chen, Wei-Ting Hung, Yin-Yi Chou, Yi-Da Wu.

**Methodology:** Hsin-Hua Chen, Ching-Heng Lin, Wen-Cheng Chao.

**Supervision:** Der-Yuan Chen.

**Writing – original draft:** Chien-Chih Chen.

**Writing – review & editing:** Hsin-Hua Chen, Chien-Chih Chen.

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
