## [Decision Letter · Decision Letter 0]

19 Jul 2019

PONE-D-19-15961

Risk of Major Autoimmune Diseases in Patients with Female Breast Cancer: a Nationwide, Population-based Cohort Study.

PLOS ONE

Dear Dr. Chien-Chih Chen,

Thank you for submitting your manuscript to PLOS ONE. After careful consideration, we feel that it has merit but does not fully meet PLOS ONE’s publication criteria as it currently stands. Therefore, we invite you to submit a revised version of the manuscript that addresses the points raised during the review process.

As you can see in the comments, both reviewers concluded that the basic data deserves to be published if certain changes, corrections and specifications have been made.  In addition, both reviewers pay attention to partly erroneous or awkward usage of English language and medical expressions. I agree with the comments and strongly encourage you to improve the text by proficient language editing and to carefully respond to the detailed comments of Reviewer 2.

We would appreciate receiving your revised manuscript by September 7, 2019. To enhance the reproducibility of your results, we recommend that if applicable you deposit your laboratory protocols in protocols.io, where a protocol can be assigned its own identifier (DOI) such that it can be cited independently in the future. For instructions see: http://journals.plos.org/plosone/s/submission-guidelines#loc-laboratory-protocols

We look forward to receiving your revised manuscript.

Kind regards,

Pirkko L. Härkönen, M.D., Ph.D.

Academic Editor

PLOS ONE

3. Thank you for stating the following in the Acknowledgments Section of your manuscript: "This study was supported by Taichung Veterans General Hospital."

 "No"

Reviewers' comments:

Reviewer's Responses to Questions

**Comments to the Author**

1. Is the manuscript technically sound, and do the data support the conclusions?

Reviewer #1: Yes

Reviewer #2: Partly

2. Has the statistical analysis been performed appropriately and rigorously? 

Reviewer #1: Yes

Reviewer #2: I Don't Know

3. Have the authors made all data underlying the findings in their manuscript fully available?

Reviewer #1: No

Reviewer #2: Yes

4. Is the manuscript presented in an intelligible fashion and written in standard English?

Reviewer #1: Yes

Reviewer #2: No

5. Review Comments to the Author

Reviewer #1: This study makes use of the large Taiwanese National Health Insurance Research Database (NHIRD) to compare the incidence rates of autoimmune diseases in breast patients and propensity score matched non-breast cancer individuals. The study design is well powered, data analysis is clean, and results are unambiguous.

I only have some minor comments—

1) In title the expression “in patients with female breast cancer” is awkward. Please consider rephrase it, for example, as “in female patients with breast cancer”?

2) There is a typo in Abstract “1:4” instead of “1:40”.

Reviewer #2: General comments

The idea is novel, and inspired by very up-to-date science: the importance of immune responses in breast cancer. In the US, there about 300 clinical studies registered on this issue. This study illustrates the potential for immunotherapy.

The paper needs English editing to give an impression of serious science. Too many typing errors that I can correct them all. I also think many sentences can be written better, for example, I think this heading is better “Risk of Major Autoimmune Diseases in female breast Cancer: a Population-based Cohort Study”. First sentence in the abstract “Breast cancer is one of common malignancy among women” illustrates the low level of proficiency in English.

Method: The method needs to be better described and justified. Why not using all the data in NHRID? Then they need not to match and later adjust for covariates. This is a case-control cohort study. Breast cancers are recorded during 2007-13, while non-breast cancer patients are recorded during the period 2003-2013. This can make problems, I think. If not, please explain. Why not use all the population in NHRID, split them in breast cancer patients and not breast cancer patients at start, and then follow them up until disease (and censor them). Then you have a prospective cohort study, and less variables to adjust for. The potential risk of biased outcome/false positive reporting in this study is related to the adjustment.

In Table 2, I would like to see estimates before adjustment of adjuvant therapy. Adjustment for CIC and adjuvant therapy may generate over-adjustment. You adjust for the same disease twice, unless you think that adjuvant therapy directly influence on the risk of autoimmune disease.

Minor comments

Abstract, Method: the ratio is 1:4, I think. You are not matching on sex, I think. You are excluding males.

Page numbers are missing. Would be fine with line numbers too.

Introduction, second sentence. Delete “prevalence”. Only focus on incidence, because women live longer than men do. Thus, comparing prevalence rates is not meaningful.

Last sentence, last paragraph, introduction (especially 6th last line): bad English, too long sentence. Furthermore, the aim/hypothesis to study has already been presented in the previous paragraph.

Data source, first paragraph, forth sentence: The word “claims” is not necessary. Sufficient to write “… comprehensive data regarding…” Last sentence: “provides data” and not “databases” to researchers.

I think “catastrophic illness” is not the proper word to use. “Possible fatal illness”, is probably better.

Definition of female breast cancer patients: CIC is not defined. And not MAD either, in the next paragraph.

Potential confounders: gender??? I suggest excluding all male breast cancer patients from start and only study women. Then adjusting for gender is not necessary.

Statistical analyses: I think you have done a case-control cohort study.

6. PLOS authors have the option to publish the peer review history of their article (what does this mean?). If published, this will include your full peer review and any attached files.

Reviewer #1: No

Reviewer #2: Yes: Per Henrik Zahl

---

## [Author Response · Author response to Decision Letter 0]

6 Aug 2019

Response to reviewers:

Dear Editors-in-Chief and Reviewers,

Reviewer #1: This study makes use of the large Taiwanese National Health Insurance Research Database (NHIRD) to compare the incidence rates of autoimmune diseases in breast patients and propensity score matched non-breast cancer individuals. The study design is well powered, data analysis is clean, and results are unambiguous.

I only have some minor comments—

1) In title the expression “in patients with female breast cancer” is awkward. Please consider rephrase it, for example, as “in female patients with breast cancer”?

Response: We had rephrased it as ‘in female breast cancer’.

2) There is a typo in Abstract “1:4” instead of “1:40”.

Response: We correct it.

Reviewer #2: General comments

General comments

The idea is novel, and inspired by very up-to-date science: the importance of immune responses in breast cancer. In the US, there about 300 clinical studies registered on this issue. This study illustrates the potential for immunotherapy.

The paper needs English editing to give an impression of serious science. Too many typing errors that I can correct them all. I also think many sentences can be written better, for example, I think this heading is better “Risk of Major Autoimmune Diseases in female breast Cancer: a Population-based Cohort Study”. First sentence in the abstract “Breast cancer is one of common malignancy among women” illustrates the low level of proficiency in English.

Response: We have rephrased the title. Changes have been made pertaining to language and grammar.

Method: The method needs to be better described and justified. Why not using all the data in NHRID? Then they need not to match and later adjust for covariates. This is a case-control cohort study. Breast cancers are recorded during 2007-13, while non-breast cancer patients are recorded during the period 2003-2013. This can make problems, I think. If not, please explain. Why not use all the population in NHRID, split them in breast cancer patients and not breast cancer patients at start, and then follow them up until disease (and censor them). Then you have a prospective cohort study, and less variables to adjust for. The potential risk of biased outcome/false positive reporting in this study is related to the adjustment.

Response: 

1. Because we were unable to get all the data in NHIRD, the only way to get matched non-breast cancer individuals was to use the one million representative population in the LHID2000. We have added a statement in Line 8 of page 6: “Because we were unable to get all the data in NHIRD, we selected the non-breast cancer comparison cohort from the population in the LHID2000.”

2. The periods of datasets for the female breast cancer cohort and the comparison cohort are the same (i.e., from 2003 to 2013). To identify new cases of breast cancer during 2007–2013, we excluded patients who had a diagnosis of breast cancer before 2007. To eliminate the differences of diagnosis and treatment in different era, the comparison cohort was also matched for the diagnosis date of female breast cancer patients.

In Table 2, I would like to see estimates before adjustment of adjuvant therapy. Adjustment for CIC and adjuvant therapy may generate over-adjustment. You adjust for the same disease twice, unless you think that adjuvant therapy directly influence on the risk of autoimmune disease.

Response: 

1. Chemotherapy and hormone therapy may affect lymphocytes and other immune cells, affecting the risk of autoimmune disease.

2. CCI was used to represent the level of general comorbid medical conditions. For female breast cancer patients, breast cancer was not included for calculating CCI. Therefore, we did not adjust for the same disease twice. We ahve added a statment in Line 8 of page 9: “For female breast cancer patients, breast cancer was not included for calculating CCI. ”

Minor comments

Abstract, Method: the ratio is 1:4, I think. You are not matching on sex, I think. You are excluding males.

Response: Since we have excluded males, we did not match the sex. We have removed “sex” in the list of matched variables.

Page numbers are missing. Would be fine with line numbers too.

Response: We have added continuous page and line numbers.

Introduction, second sentence. Delete “prevalence”. Only focus on incidence, because women live longer than men do. Thus, comparing prevalence rates is not meaningful.

Response: We have corrected it according to your suggestion.

Last sentence, last paragraph, introduction (especially 6th last line): bad English, too long sentence. Furthermore, the aim/hypothesis to study has already been presented in the previous paragraph. 

Response: We have revised the statement for concision a follows. “The aim of this study was to investigate the risks of major autoimmune diseases in female breast cancer patients using the NHIRD.” We have deleted a redundant statement regarding the hypothesis.

Data source, first paragraph, forth sentence: The word “claims” is not necessary. Sufficient to write “… comprehensive data regarding…” Last sentence: “provides data” and not “databases” to researchers.

Response: We have corrected it based on your suggestions.

I think “catastrophic illness” is not the proper word to use. “Possible fatal illness”, is probably better.

Response: We have used “possible fatal illness” to describe the characteristics of these diseases. However, the official name of the certificate for patients with these “possible fatal illness” is “catastrophic illness certificate” in Taiwan.

Definition of female breast cancer patients: CIC is not defined. And not MAD either, in the next paragraph.

Response: 

1. To avoid confusion with CCI, we have removed the abbreviation of CIC and used “catastrophic illness certificate” in the manuscript. 

2. Because MAD is not a common abbreviation, we have deleted this abbreviation and used the entire definition “major autoimmune disease” in manuscript.

Potential confounders: gender??? I suggest excluding all male breast cancer patients from start and only study women. Then adjusting for gender is not necessary.

Response: Because we have excluded males, we have removed “gender” in the list of potential confounders.

Statistical analyses: I think you have done a case-control cohort study. 

Response: This study is a retrospective cohort study and not case-control study. Therefore, we have used the Cox regression analysis.

---

## [Decision Letter · Decision Letter 1]

4 Sep 2019

[EXSCINDED]

PONE-D-19-15961R1

Risk of Major Autoimmune Diseases in Female Breast Cancer Patients: A Nationwide, Population-based Cohort Study

PLOS ONE

Dear Dr. Chien-Chih Chen,

Thank you for submitting your manuscript to PLOS ONE. After careful consideration, we feel that it has merit but does not fully meet PLOS ONE’s publication criteria as it currently stands. Therefore, we invite you to submit a revised version of the manuscript that addresses the points raised during the review process.

Although the results of your study are considered important and the analyses basically appropriate, the  presentation of the results of research on medical topics would benefit from revision. Please carefully consider the detailed and justified comments of the Reviewer. The revision of the text to them would clearly help making your results more understandable and convincing to the interested readers.

We would appreciate receiving your revised manuscript by September 15th, 2019. To enhance the reproducibility of your results, we recommend that if applicable you deposit your laboratory protocols in protocols.io, where a protocol can be assigned its own identifier (DOI) such that it can be cited independently in the future. For instructions see: http://journals.plos.org/plosone/s/submission-guidelines#loc-laboratory-protocols

We look forward to receiving your revised manuscript.

Kind regards,

Pirkko L. Härkönen, M.D., Ph.D.

Academic Editor

PLOS ONE

Reviewers' comments:

Reviewer's Responses to Questions

**Comments to the Author**

1. If the authors have adequately addressed your comments raised in a previous round of review and you feel that this manuscript is now acceptable for publication, you may indicate that here to bypass the “Comments to the Author” section, enter your conflict of interest statement in the “Confidential to Editor” section, and submit your "Accept" recommendation.

Reviewer #2: All comments have been addressed

2. Is the manuscript technically sound, and do the data support the conclusions?

Reviewer #2: Partly

3. Has the statistical analysis been performed appropriately and rigorously? 

Reviewer #2: Yes

4. Have the authors made all data underlying the findings in their manuscript fully available?

Reviewer #2: (No Response)

5. Is the manuscript presented in an intelligible fashion and written in standard English?

Reviewer #2: No

6. Review Comments to the Author

Reviewer #2: Abstract, lines 12-14: ….selected females diagnosed with breast cancer in the period 2007 to 2013 into a control group.

Abstract, line 15. Delete sex. Not necessary to match on sex when you only study women.

Abstract, lines 19-21. ….For SLE, the incidence rates were 2.3 (BC group) vs 10.0 (control group) per 100,000 women year; for RA rates were 19.3 (BC group) vs 42.7 (control group) per 100,000 women year; and for SS rates were 20.5 (BC group) vs 38.2 (control group) per 100,000 women year.

Do not use 10 raised to the power of fifth in a medical journal.

Abstract, page 3 , lines 1-4: After adjusting for potential confounders, the hazard ratio (HZ) with corresponding confidence interval (CI) for females breast cancer patients vs control group were 0.04 (95% CI: 0.01-0.24) for SLE; 0.03 (95% CI: 0.02-0.04) for RA; and 0.21 (95% CI: 0.09-0.48) for SS.

Page 4, line 2: ….one of the most common….

Page 5, lines 13-14: The data set were extracted from the NHIRD database using data from the period 2003 to 2013.

Page 6, lines 5 -6: …..,including data on both inpatients and outpatients,….

Page 6, lines 9-10:….,we selected the non-breast cancer control group from the LHID cohort.

Page 7, line 13: delete leaving and write “emigration”. Is there any other way of leaving the database than emigration or death?

Page 8, line 6: …,as the control group.

Page 8, line 10: including 409,340 female non-breast cancer individuals before matching.

Page 9, line 17…,by using Student’s t-test,….by using Pearson’s…

Page 9, line18; confidence interval has been defined above.

Page 10, line 2: HZ has been defined before.

Page 10, lines 13-14: …. ,the incidence rate of SLE was 2.3 per 100,000 women years, the incidence rate of RA was 19.3 per 100,000 women years, the incidence rate of SS was 20.1 per 100,000 women years and the incidence rate of DMtis/PM was 2.3 per 100,000 women years.

Page 10, lines 15-16: Do as above.

Page 10, line 19: , the hazard ratio (HZ) with corresponding confidence interval (CI) for females breast cancer patients vs control group were 0.04 (95% CI: 0.01-0.24) for SLE; 0.03 (95% CI: 0.02-0.04) for RA; and 0.21 (95% CI: 0.09-0.48) for SS.

Page 12, line 5: …had a significant lower risk….

Page 13, line 6. Do not use the word “claim data”. This is register data.

Page 15, Table 1. Spelling error in the line under “Concomitant medication”. FiRst

Page 15, Table 1: “Time to first SLE”… The number look like real numbers with percentage in parenthesis and not life times. Please correct or explain.

7. PLOS authors have the option to publish the peer review history of their article (what does this mean?). If published, this will include your full peer review and any attached files.

Reviewer #2: Yes: Per-Henrik Zahl

---

## [Author Response · Author response to Decision Letter 1]

6 Sep 2019

Dear Editors-in-Chief and Reviewers,

Reviewer #2: 

Abstract, lines 12-14: ….selected females diagnosed with breast cancer in the period 2007 to 2013 into a control group.

Response: We rephrased abstract. We identified newly-diagnosed female breast cancer patients and randomly selected females without breast cancer in the period 2007 to 2013 into a control group.

Abstract, line 15. Delete sex. Not necessary to match on sex when you only study women.

Response: We have deleted ‘sex’.

Abstract, lines 19-21. ….For SLE, the incidence rates were 2.3 (BC group) vs 10.0 (control group) per 100,000 women year; for RA rates were 19.3 (BC group) vs 42.7 (control group) per 100,000 women year; and for SS rates were 20.5 (BC group) vs 38.2 (control group) per 100,000 women year. Do not use 10 raised to the power of fifth in a medical journal.

Response: We rephrased abstract.

Abstract, page 3 , lines 1-4: After adjusting for potential confounders, the hazard ratio (HZ) with corresponding confidence interval (CI) for females breast cancer patients vs control group were 0.04 (95% CI: 0.01-0.24) for SLE; 0.03 (95% CI: 0.02-0.04) for RA; and 0.21 (95% CI: 0.09-0.48) for SS.

Response: We rephrased abstract: After adjusting for potential confounders, the hazard ratios (95% confidence intervals) for female breast cancer patients vs. control group were 0.04 (0.01–0.24) for SLE; 0.03 (0.02–0.04) for RA; and 0.21 (0.09–0.48) for SS.

Page 4, line 2: ….one of the most common….

Response: We rephrased introduction: Breast cancer is one of the most common malignancies among women.

Page 5, lines 13-14: The data set were extracted from the NHIRD database using data from the period 2003 to 2013.

Response: We rephrased method.

Page 6, lines 5 -6: …..,including data on both inpatients and outpatients,….

Response: We rephrased method.

Page 6, lines 9-10:….,we selected the non-breast cancer control group from the LHID cohort.

Response: We rephrased method.

Page 7, line 13: delete leaving and write “emigration”. Is there any other way of leaving the database than emigration or death?

Response: We rephrased method. We defined the censor date as December 31, 2013 (the last date of the database used) or the date of withdrawal from the NHI for emigration or death.

Page 8, line 6: …,as the control group.

Response: We rephrased method.

Page 8, line 10: including 409,340 female non-breast cancer individuals before matching.

Response: We rephrased method.

Page 9, line 17…,by using Student’s t-test,….by using Pearson’s…

Response: We rephrased method.

Page 9, line18; confidence interval has been defined above.

Response: Because CI was not defined before in the revised manuscript, so we kept the definition of HZ here.

Page 10, line 2: HZ has been defined before.

Response: Because HZ was not defined before in the revised manuscript, so we kept the definition of HZ here.

Page 10, lines 13-14: …. ,the incidence rate of SLE was 2.3 per 100,000 women years, the incidence rate of RA was 19.3 per 100,000 women years, the incidence rate of SS was 20.1 per 100,000 women years and the incidence rate of DMtis/PM was 2.3 per 100,000 women years.

Response: We rephrased results.

Page 10, lines 15-16: Do as above.

Response: We rephrased results.

Page 10, line 19: , the hazard ratio (HZ) with corresponding confidence interval (CI) for females breast cancer patients vs control group were 0.04 (95% CI: 0.01-0.24) for SLE; 0.03 (95% CI: 0.02-0.04) for RA; and 0.21 (95% CI: 0.09-0.48) for SS.

Response: We rephrased results.

Page 12, line 5: …had a significant lower risk….

Response: We rephrased discussion.

Page 13, line 6. Do not use the word “claim data”. This is register data.

Response: We rephrased discussion.

Page 15, Table 1. Spelling error in the line under “Concomitant medication”. FiRst

Response: We correct spelling error.

Page 15, Table 1: “Time to first SLE”… The number look like real numbers with percentage in parenthesis and not life times. Please correct or explain.

Response: We correct Table 1.

Thank you for your helpful suggestion.

---

## [Editor Report · Decision Letter 2]

10 Sep 2019

Risk of Major Autoimmune Diseases in Female Breast Cancer Patients: A Nationwide, Population-based Cohort Study

PONE-D-19-15961R2

Dear Dr. Chien-Chih Chen,

We are pleased to inform you that your manuscript has been judged scientifically suitable for publication and will be formally accepted for publication once it complies with all outstanding technical requirements.

With kind regards,

Pirkko L. Härkönen, M.D., Ph.D.

Academic Editor

PLOS ONE
---

## [Editor Report · Acceptance letter]

12 Sep 2019

PONE-D-19-15961R2 

Risk of Major Autoimmune Diseases in Female Breast Cancer Patients: A Nationwide, Population-based Cohort Study 

Dear Dr. Chen:

I am pleased to inform you that your manuscript has been deemed suitable for publication in PLOS ONE. Congratulations! Your manuscript is now with our production department. 

With kind regards,

on behalf of

Dr. Pirkko L. Härkönen 

Academic Editor

PLOS ONE